

# Analysis and forecasts for trends of COVID-19 in Pakistan using Bayesian models

Navid Feroze[1], Kamran Abbas[1], Farzana Noor[2] and Amjad Ali[3]

[1] Department of Statistics, The University of Azad Jammu and Kashmir, Muzaffarabad, Pakistan
[2] Department of Mathematics and Statistics, International Islamic University, Islamabad, Pakistan
[3] Department of Statistics, Islamia College, Peshawar, Khyber Pakhtunkhwa, Pakistan

## ABSTRACT

**Background:** COVID-19 is currently on full flow in Pakistan. Given the health facilities in the country, there are serious threats in the upcoming months which could be very testing for all the stakeholders. Therefore, there is a need to analyze and forecast the trends of COVID-19 in Pakistan.
**Methods:** We have analyzed and forecasted the patterns of this pandemic in the country, for next 30 days, using Bayesian structural time series models. The causal impacts of lifting lockdown have also been investigated using intervention analysis under Bayesian structural time series models. The forecasting accuracy of the proposed models has been compared with frequently used autoregressive integrated moving average models. The validity of the proposed model has been investigated using similar datasets from neighboring countries including Iran and India.
**Results:** We observed the improved forecasting accuracy of Bayesian structural time series models as compared to frequently used autoregressive integrated moving average models. As far as the forecasts are concerned, on August 10, 2020, the country is expected to have 333,308 positive cases with 95% prediction interval [275,034–391,077]. Similarly, the number of deaths in the country is expected to reach 7,187 [5,978–8,390] and recoveries may grow to 279,602 [208,420–295,740]. The lifting of lockdown has caused an absolute increase of 98,768 confirmed cases with 95% interval [85,544–111,018], during the post-lockdown period. The positive aspect of the forecasts is that the number of active cases is expected to decrease to 63,706 [18,614–95,337], on August 10, 2020. This is the time for the concerned authorities to further restrict the active cases so that the recession of the outbreak continues in the next month.

## INTRODUCTION

During December, 2019, the historical pandemic started in Wuhan, China (*Paules, Marston & Fauci, 2020*). This pandemic was named as a novel coronavirus COVID-19 by the World Health Organization (*WHO, 2020*). Though this virus has lower death rates when compared to Middle East Respiratory Syndrome (MERS) and Severe Acute Respiratory Syndrome (SARS), this virus has higher transmission rates

Corresponding author
Navid Feroze,
navidferoz@gmail.com

(*Tomar & Gupta, 2020*). Due to its higher transmissibility, the virus has covered almost the whole world (*WHO, 2020*). In Pakistan, the first positive case was observed on February 26, 2020. The country imposed a complete lockdown on March 23, 2020. However, the struggling economy of the country forced the government to lift the lockdown on May 9, 2020. During the lockdown, the number of new cases was low (*Yousaf et al., 2020*); however, the outbreak has increased pace since lifting of the lockdown. There are further expectations that the number of cases and deaths will increase more rapidly in the future (*Yousaf et al., 2020*).

The repeated short term forecasts for the patterns of the pandemic are fundamental (*Li et al., 2020*; *Wang & Zhang, 2020*; *Zhou et al., 2020*). These short term forecasts help the policy makers to make informed decisions in accordance with the evolving scenarios (*Ippolito et al., 2020*; *McCloskey et al., 2020*). There are a number of contributions regarding the short terms forecasts for the various parameters of the pandemic in different countries. Some important studies have been conducted in China (*Li et al., 2020*; *Perc et al., 2020*; *Fanelli & Piazza, 2020*), France (*Fanelli & Piazza, 2020*), Germany (*Perc et al., 2020*), India (*Tomar & Gupta, 2020*; *Gupta & Pal, 2020*), Iran (*Perc et al., 2020*; *Zhan et al., 2020*; *Moftakhar, Seif & Safe, 2020*), Italy (*Fanelli & Piazza, 2020*; *Zhan et al., 2020*), Nigeria (*Majeed, Adeleke & Popoola, 2020*), South Korea (*Zhan et al., 2020*) and the United States (*Perc et al., 2020*). The literature contains lots of models to obtain forecasts/predictions in different situations; machine learning (*Ou, 2016*; *Le, 2019*; *Liu et al., 2020*), deep learning (*Le, Ho & Ou, 2017*; *Le et al., 2019*), ARIMA models (*Majeed, Adeleke & Popoola, 2020*; *Benvenuto et al., 2020*; *Zhang et al., 2020*) and Neural Network (*Moftakhar, Seif & Safe, 2020*).

There has been an earlier attempt to forecast the number of cases, recoveries and deaths in Pakistan (*Yousaf et al., 2020*; *Khan, Saeed & Ali, 2020*). However, the contribution by *Yousaf et al. (2020)* was based on quite smaller datasets which is serious issue for producing reliable predictions (*Moftakhar, Seif & Safe, 2020*). Furthermore, these forecasts underestimated the number of confirmed cases and deaths which may be due to changing post-lockdown trends in the country. On the other hand, the contribution by *Khan, Saeed & Ali (2020)* did not considered the causal impacts of the lockdown in the country. In addition, the said contributions used the ARIMA models (*Yousaf et al., 2020*) and vector autoregressive models (*Khan, Saeed & Ali, 2020*) for the forecasts. The ARIMA models have been frequently used to forecast the patterns of this pandemic (*Gupta & Pal, 2020*; *Majeed, Adeleke & Popoola, 2020*; *Benvenuto et al., 2020*; *Kumar et al., 2020*). However, these models have some limitations. Firstly, the forecasts from these models are dependent on the previous behavior of the data along with preceding forecast errors, which means the forecasting error accumulates over time. In addition, the forecasting accuracy of these models is affected in presence of covariates (*Brockwell & Davis, 2002*). To avoid such issues, the Bayesian structural time series (BSTS) models are employed (*Scott & Varian, 2013*; *Brodersen et al., 2015*). The BSTS models (i) allow the inclusion of prior information (ii) allow the model parameters to evolve over time (iii) can handle large number of

covariates using spike and slab prior (iv) are less dependent on certain hypothesized specifications (v) can investigate individual components of a time series (*McQuire et al., 2019*). Interestingly, number of time series models, such as ARIMA (used by *Yousaf et al., 2020*) and vector autoregressive models (used by *Khan, Saeed & Ali, 2020*), can be expressed in the state space form (*Scott & Varian, 2014b*). These models have the capability to provide reliable forecasts for future outbreak of different diseases (*Scott & Varian, 2014a*). The said models have been used earlier in forecasting the health issues by using alcohol and alcohol licensing policies (*de Vocht et al., 2017*; *McQuire et al., 2019*). The proposed models provided 14% improvement (while comparing with ARIMA models) in forecasting the consumer sentiments (*Scott & Varian, 2014a*).

The strict social distancing in Wuhan facilitated China to restrict the spread of the pandemic in other provinces (*Li et al., 2020*). However, in Pakistan the lockdown was lifted at a quite earlier stage of the pandemic. Unfortunately, to the best of our knowledge, the effects of ending the lockdown have not been studied yet. Therefore, the analysis of the impacts of lifting the lockdown in the country is very important. The causal impacts of different interventions can also be computed using BSTS models. Unlike ARIMA models, these models compute the dynamic confidence interval for the evolving impact, based on difference between actual and counterfactual series. Due to their added advantages, these models are superior to conventional models (*Brodersen et al., 2015*).

We have conducted a study to investigate the various parameters of COVID-19 in Pakistan, for next 30 days. Since the Bayesian methods often produce better results as compared classical methods (*Kundu & Joarder, 2006*; *Kundu, 2008*; *Pak, Parham & Saraj, 2013*), the BSTS models have been proposed for the forecasting. The causal impacts of lifting the lockdown in the country have also been investigated conducting intervention analysis within BSTS models. The improved forecast accuracy has been observed using BSTS models, instead of repeatedly used ARIMA models. We have also included the datasets from neighboring countries (India and Iran) to investigate the performance of the proposed models for data regarding other countries. The detailed study suggested that the cumulative number of confirmed cases and deaths is expected to increase exponentially during next month. However, the recoveries are expected to increase faster than confirmed cases, so the active number of cases will decrease during the next month. The study also revealed that lifting the lockdown at the earlier stage has seriously increased the trajectory of the outbreak in the country.

## MATERIALS AND METHODS

The data have been obtained from the published reports of the National Institute of Health (NIH), Islamabad, Pakistan. The data contain the details regarding cumulative (and daily) number of cases, deaths, recoveries and tests in the country (*NIH, 2020*). NIH updates the data on daily basis since February 26, 2020. Additionally, data for Iran and India are obtained from reports of *Our World in Data (2020)*. As we have used the published data by NIH and Our World in Data, ethical approval was not required.

## ARIMA models

The ARIMA model is mathematically written as ARIMA (p, d, q). The model is based on three parameters 'p, d and q'. The parameter 'p' represents the autoregressive terms, 'd' defines the number of non-seasonal differences and 'q' denotes the number of moving terms. The ARIMA (p, d, q) model can be written as

$$Z_t = \theta_1 z_{t-1} + \theta_2 z_{t-2} + \ldots + \theta_p z_{t-p} + \varepsilon_1 - \lambda_1 \varepsilon_{t-1} - \varepsilon_2 - \lambda_2 \varepsilon_{t-2} - \ldots - \varepsilon_q - \lambda_q \varepsilon_{t-q} \quad (1)$$

where, $\theta_p$ represents the terms of autoregressive operator, $\varepsilon_q$ are the coefficients of the error terms, $\lambda_q$ are the values of moving average operator and $Z_t$ is d-order differenced time series.

## Structural time series models

Visualization of a time series as a product of aggregating different components is very useful. The decomposition of each layer facilitates the direct individual interpretation of the model. The structural time series models have these features. The basic form of the structural time series models (STM) is as follows:

$$Y_t = T_t + S_t + E_t$$

where $Y_t$ is the observed value, $T_t$ represents the trend component, $S_t$ denotes the seasonal component, and $E_t$ is the error term. Hence, the STM is a dynamic system composed of trend and seasonal components perturbed by random errors. The STM is capable of reflecting the important characteristics of the data and enhancing prediction power of the model by including repressors. These models also utilize the previous knowledge about the parameters by adding the prior information.

The Gaussian STM can be expressed as

$$Y_t = W_t \lambda_t + \omega_t; \ \omega_t \sim N(0, \Phi_t) \tag{2}$$

$$\lambda_{t+1} = X_t \lambda_t + H_t \mu_t; \ \mu_t \sim N(0, \Omega_t) \tag{3}$$

Equation (2) links the observed data $Y_t$ with the unobserved latent variable $\lambda_t$, hence called observation equation. In (2), $Y_t$ is $k \times 1$ vector of values, $W_t$ is a $k \times m$ matrix involving known values, $\lambda_t$ is unobserved $k \times 1$ state vector and $\omega_t$ are the randomly and independently distributed Gaussian error terms with zero mean and variance $\Phi_t$. On the other hand, Eq. (3) is called transition equation as it defines the behavior of latent state over time. It is simply an autoregressive model of $\lambda_t$, defined by the unobservable Markov Chain process observed by $Y_t$. Here, the $X_t$ is $k \times k$ transition matrix, $H_t$ is $k \times m$ error control matrix (indentifies the rows of transition equation having non-zero

error terms), and $\mu_t$ another Gaussian random error term with mean zero and variance $\Omega_t$. Therefore, the STM contain the underlying stochastic process determined by $\lambda_t$. Since $\lambda_t$ are unobservable, a vector of observations is used to compute the system. The initial information in shape of prior information is assumed for $\lambda$ which is normally distributed with mean $\lambda_t$ and variance $Q_0$ which is independent of $\omega_t$ and $\mu_t$ for all t.

## Markov Chain Monte Carlo (MCMC) method

Suppose $\beta$ is a set of all model parameters and $\alpha = (\alpha_1, \alpha_2, \ldots, \alpha_m)$ represents the full state sequence. Further suppose that the prior distribution for $\beta$ is $\pi(\beta)$ as well as distribution $\pi(\alpha_0|\beta)$ on the initial state values. Then, the sample from the posterior distribution $p(\alpha, \beta|\mathbf{Y})$ can be obtained using MCMC method in the following manner. Draw a sequence of $(\alpha_1, \beta_1), (\alpha_1, \beta_1), \ldots$ from a Markov chain having stationary distribution as $p(\alpha, \beta|\mathbf{Y}_{1:n})$. The sampler alternates between the data augmentation step and parameter simulation step. The data augmentation step and parameter estimation step simulate from conditional distributions $p(\alpha|\mathbf{Y}_{1:n}, \beta)$ and $p(\beta|\mathbf{Y}_{1:n}, \alpha)$, respectively. The data augmentation step has been carried out using the algorithm proposed by *Durbin & Koopman (2002)* which provides the improvement in the algorithm proposed by *De Jong & Shephard (1995)*. Once the draws for the state are available, the draws for the parameters are easy to obtain for all state components excluding the static regression coefficients. The draws for the static regression coefficients have been obtained using the algorithm proposed by *Ghosh & Clyde (2011)*. The detailed studies regarding implementation of MCMC algorithms can be seen from the contributions of *Fitzgerald (2001)*, *Bugallo, Martino & Corander (2015)* and *Martino (2018)*.

## Causal impacts

Equations (2) and (3) also investigate the post-lockdown discrepancy between observed number of new cases, deaths and active cases and a simulation based series that was expected without lifting the lockdown. Such investigations provide facilitations in assessing the causal impacts of lifting lockdown in the country. These causal impacts can be computed by the following steps.

Step-I: Fit STM using observations under the lockdown period.

Step-II: Use fitted model from Step-I to produce forecasts for post-lockdown period.

Step-III: The difference between forecasted series and actual series during the post-lockdown period is computed and interpreted as causal impacts of lifting the lockdown in the country.

The causal impact model incorporates the behavior of two time series (i) the behavior of the response series in the pre-intervention period (ii) the behavior of a time series that were predictive of the response series in the pre-intervention period. If the control series did not encountered the intervention, it can be assumed that the relationship that existed between the treatment and control in the pre-intervention period may continue in the

post-intervention period (*Brodersen et al., 2015*). We used number tests as the control series, which were predictive of the number of cases/deaths prior to the lifting of the lockdown. The number of tests was itself not affected by the lifting of the lockdown; hence the assumptions for the implementation of the causal impact model were not affected.

We have used R software to conduct the analysis regarding the study. The computations for ARIMA models have been obtained using auto.arima function available in forecast package, while bsts package has been used to obtain the forecasts for the BSTS models (*Scott, 2020*). We have employed the local level model for the trend component. The number seasons have been chosen to be seven to evaluate the weekly pattern of the series. The root mean square percentage error (RMSPE) and root median square percentage error (RMdSPE) have been used to compare the forecast accuracy of proposed models with ARIMA models. The said measures can efficiently determine the forecast accuracy of a model (*Hyndman & Koehler, 2006*; *Bowerman, O'Connell & Koehler, 2004*). Based on these measures, we have investigated the predictive power of BSTS models using different starting points. To investigate whether the residuals are white noise, we used Ljung Box test at different lags. We have also included the datasets from neighboring countries (India and Iran) to observe the applicability of the proposed models in different situations. As we observed the improved forecasting capacity of BSTS models, the detailed forecasts have only been reported under BSTS models for brevity. The forecasts under BSTS models are based on prior information and current data (likelihood function). The bsts package uses spike-slab prior for the analysis. The prior information is combined with the likelihood function to produce the posterior distribution. The BSTS models often use the default prior. The default prior for the variance ($\sigma^2$) can be formulated as: $\sigma^{-2} \sim G\left(10^{-2}, 10^{-2} s_Y^2\right)$, where $G(a, b)$ is gamma distribution with parameters 'a' and 'b' and $s_Y^2 = \sum_t \left(Y_t - \bar{Y}\right)^2 / (n - 1)$. On the other hand in case of several controls, the spike and slab prior is used for the coefficients (*Scott & Varian, 2014b*). The prior parameters in spike and slab are selected using Zellner's prior method (*Liang et al., 2008*; *Zellner, 1986*).The posterior distribution is estimated using Gibbs sampler (*George & McCulloch, 1997*). The detailed studies regarding implementation of MCMC algorithms can be seen from the contributions of (*Fitzgerald, 2001*; *Bugallo, Martino & Corander, 2015*; *Martino, 2018*). In addition the Kalman filter and Bayesian model averaging were employed to obtain the forecasts. The causal impacts of lifting the lockdown are investigated by conducting intervention analysis within the BSTS models. The CausalImpact package in R (*Brodersen & Hauser, 2020*) has been used for estimating the said impacts.

## RESULTS AND DISCUSSIONS

On July 11, 2020, Pakistan had 246,351 number of total confirmed cases, 93,217 active cases, 5,123 deaths and 153,134 recoveries regarding COVID-19. The ratio of cumulative confirmed cases per 100 tests (RCT) stood at 16.01. Similarly, the ratio of cumulative

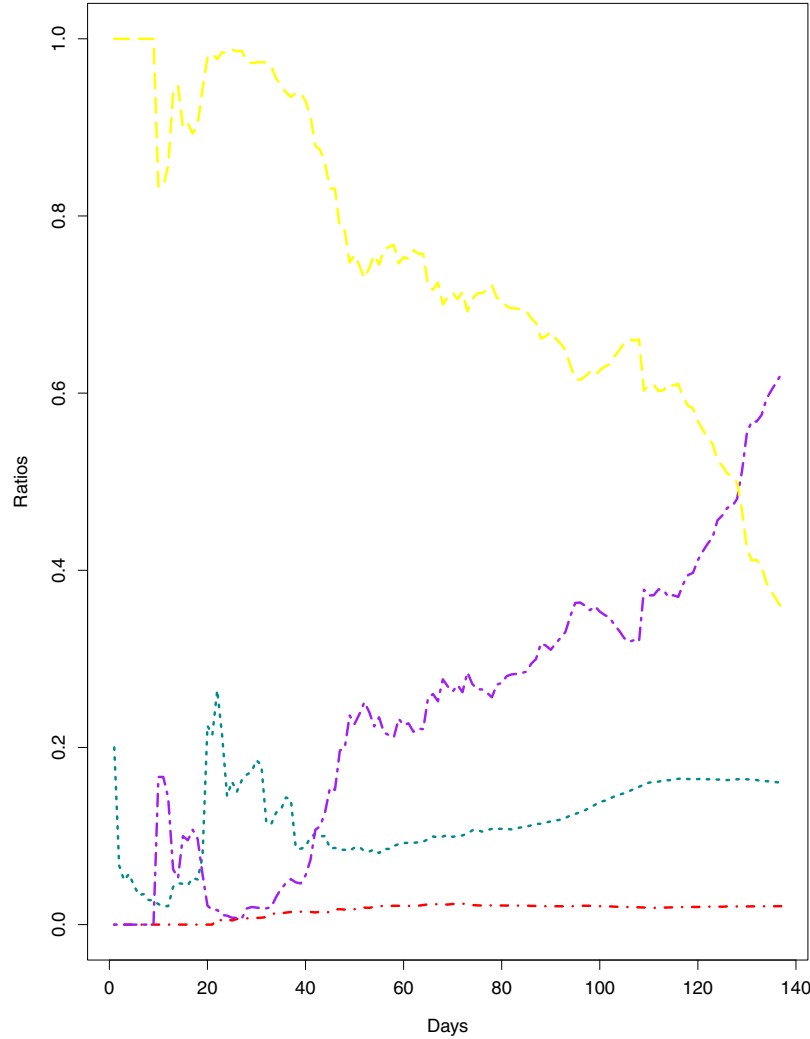

**Figure 1 Current Trends of the outbreak.** The data points represent the ratios for different parameters of the pandemic. The green line indicates the Ratio of cumulative number of cases to cumulative number of tests (RCT). The blue line indicates the ratio of cumulative number of deaths to the cumulative number of cases (RDC). The red line shows the ratio of cumulative number of recoveries to the cumulative number of cases (RRC). The yellow line represents the ratio of cumulative number of active cases to the cumulative number of confirmed cases (RAC).     

deaths per 100 confirmed cases (RDC) was 2.08, and the ratio of cumulative number of recoveries per 100 confirmed cases (RRC) was 62.12. These figures have already challenged the healthcare system of Pakistan having total number of 215,436 doctors, 108,137 nurses, 19,218 community health workers (CHWs) and 41,689 labs across the country (*WHO, 2020*). The government figures also suggest that there are 6,664 crucial care beds and about 2,500 Ventilators for the COVID-19 patients in the country (*NIH, 2020*). These figures simply indicate that Pakistan has very limited healthcare infrastructure to cope up this pandemic. Hence, analysis of different parameters of this epidemic is fundamental for watchful decision making in the country. Using the dataset obtained from NIH, we compared the forecasting accuracy of BSTS models with

| Country/parameter | ARIMA | | BSTS | | ARIMA | | BSTS | |
|---|---|---|---|---|---|---|---|---|
| | RMSPE | RMdSPE | RMSPE | RMdSPE | RMSPE | RMdSPE | RMSPE | RMdSPE |
| Pakistan | 90% of the series | | | | 80% of the series | | | |
| Cases | 0.0792 | 0.0122 | 0.0708 | 0.0093 | 0.0230 | 0.0110 | 0.0243 | 0.0064 |
| Deaths | 0.0581 | 0.0104 | 0.0569 | 0.0095 | 0.0440 | 0.0103 | 0.0479 | 0.0087 |
| Recoveries | 0.1185 | 0.0240 | 0.111 | 0.0266 | 0.0799 | 0.0192 | 0.0844 | 0.0230 |
| Active | 0.0831 | 0.0173 | 0.0827 | 0.0187 | 0.0306 | 0.0143 | 0.0304 | 0.0168 |
| Pakistan | 70% of the series | | | | 60% of the series | | | |
| Cases | 0.0143 | 0.0090 | 0.0104 | 0.0055 | 0.0186 | 0.0108 | 0.0171 | 0.0062 |
| Deaths | 0.0175 | 0.0081 | 0.0153 | 0.0062 | 0.0333 | 0.0094 | 0.0316 | 0.0079 |
| Recoveries | 0.0396 | 0.0138 | 0.0409 | 0.0135 | 0.0530 | 0.0162 | 0.0475 | 0.0150 |
| Active | 0.0241 | 0.0126 | 0.0233 | 0.0128 | 0.0308 | 0.0132 | 0.0292 | 0.0169 |
| Iran | 90% of the series | | | | 80% of the series | | | |
| Cases | 0.0162 | 0.0035 | 0.0164 | 0.0017 | 0.0151 | 0.0030 | 0.0154 | 0.0016 |
| Deaths | 0.0191 | 0.0025 | 0.0158 | 0.0019 | 0.0073 | 0.0023 | 0.0073 | 0.0016 |
| Recoveries | 0.0537 | 0.0028 | 0.0206 | 0.0015 | 0.0268 | 0.0025 | 0.0178 | 0.0015 |
| Active | 0.0426 | 0.0175 | 0.0316 | 0.0171 | 0.0324 | 0.0138 | 0.0178 | 0.0131 |
| Iran | 70% of the series | | | | 60% of the series | | | |
| Cases | 0.0034 | 0.0020 | 0.0041 | 0.0017 | 0.0050 | 0.0023 | 0.0046 | 0.0012 |
| Deaths | 0.0030 | 0.0017 | 0.0025 | 0.0009 | 0.0040 | 0.0020 | 0.0028 | 0.0011 |
| Recoveries | 0.0037 | 0.0017 | 0.0019 | 0.0011 | 0.0145 | 0.0022 | 0.0054 | 0.0012 |
| Active | 0.0202 | 0.0115 | 0.0142 | 0.0105 | 0.0285 | 0.0132 | 0.0154 | 0.0112 |
| India | 90% of the series | | | | 80% of the series | | | |
| Cases | 0.0594 | 0.0103 | 0.0577 | 0.0040 | 0.0408 | 0.0088 | 0.0333 | 0.0037 |
| Deaths | 0.0463 | 0.0098 | 0.0445 | 0.0097 | 0.0393 | 0.0096 | 0.0393 | 0.0083 |
| Recoveries | 0.0702 | 0.0176 | 0.0634 | 0.0115 | 0.0656 | 0.0167 | 0.0584 | 0.0102 |
| Active | 0.0600 | 0.0131 | 0.0495 | 0.0117 | 0.0431 | 0.0126 | 0.0376 | 0.0108 |
| India | 70% of the series | | | | 60% of the series | | | |
| Cases | 0.0113 | 0.0068 | 0.0071 | 0.0018 | 0.0164 | 0.0074 | 0.0113 | 0.0024 |
| Deaths | 0.0225 | 0.0068 | 0.0213 | 0.0065 | 0.0220 | 0.0080 | 0.0213 | 0.0072 |
| Recoveries | 0.0215 | 0.0095 | 0.0183 | 0.0072 | 0.0388 | 0.0119 | 0.0293 | 0.0087 |
| Active | 0.0198 | 0.0121 | 0.0181 | 0.0090 | 0.0218 | 0.0122 | 0.0204 | 0.0102 |

**Table 1  Measures of forecast accuracy for BSTS models and ARIMA models.**

ARIMA models. The comparison is made on the basis of RMSPE and RMdSPE.

The current trends of the pandemic have been represented in Fig. 1. The results regarding comparison of predictive powers of the proposed models are given in Table 1.

The diagnostic checking for the proposed models, using Ljung Box test, has been reported in Table 2. The future trends of the outbreak are presented in Table 3 and in Fig. 2. The separated analysis of trend, seasonality and regression components has been presented

**Table 2 Results of the Ljung Box test (Q-statistic) for diagnostic checking of the proposed models.**

| BSTS Model for | Lag-5 | | Lag-10 | | Lag-30 | |
|---|---|---|---|---|---|---|
| | Q-Statistic | P-value | Q-Statistic | P-value | Q-Statistic | P-value |
| Total cases | 2.297 | 0.807 | 6.478 | 0.774 | 28.205 | 0.560 |
| Deaths | 10.088 | 0.073 | 12.414 | 0.258 | 42.046 | 0.071 |
| Recoveries | 3.108 | 0.683 | 3.750 | 0.958 | 11.894 | 0.999 |
| Active cases | 7.658 | 0.176 | 11.639 | 0.310 | 32.847 | 0.411 |

**Table 3 Summary of forecasts and expected required resources after 1 month (on August 10, 2020).**

| Item | Point forecast | 95% Prediction interval | | RMSPE | RMdSPE |
|---|---|---|---|---|---|
| Pakistan | | | | | |
| No. of expected cases | 333,308 | 275,034 | 391,077 | 0.0953 | 0.0128 |
| No. of expected deaths | 7,187 | 5,978 | 8,390 | 0.0670 | 0.0116 |
| No. of expected recoveries | 279,602 | 208,420 | 345,740 | 0.1033 | 0.0259 |
| No. of active cases | 63,706 | 19,614 | 95,337 | 0.0966 | 0.0196 |
| Iran | | | | | |
| No. of expected cases | 325,391 | 283,403 | 369,613 | 0.0176 | 0.0035 |
| No. of expected deaths | 17,458 | 15,538 | 19,476 | 0.0370 | 0.0026 |
| No. of expected recoveries | 294,493 | 243,470 | 350,203 | 0.0227 | 0.0023 |
| No. of active cases | 5,892 | 973 | 73,416 | 0.0462 | 0.0183 |
| India | | | | | |
| No. of expected cases | 1,874,037 | 1,538,452 | 2,393,316 | 0.0690 | 0.0132 |
| No. of expected deaths | 39,464 | 30,407 | 48,765 | 0.0554 | 0.0098 |
| No. of expected recoveries | 1,299,328 | 1,091,762 | 1,618,996 | 0.0790 | 0.0236 |
| No. of active cases | 517,153 | 390,286 | 717,715 | 0.0665 | 0.0167 |

in Fig. 3. On the other hand, the causal impact of lifting lockdown in the country has been discussed in Fig. 4.

Figure 1 reports the current trajectories of COVID-19 in Pakistan, where the zeroth day denotes February 25, 2020. In particular, this figure includes the trends of RCT, RDC, RRC and RAC. Though RAC has decreasing trend and RRC has increasing trend, the issue for the country is that RDC is constant over time, which means that rate of deaths is directly proportional to number of cases. In addition, the RCT has increased over time, which is quit alarming. It means the relative pace of spread has increased over time. The comparison of forecast accuracy for ARIMA and BSTS models are placed in Table 1. The numerical results for RMSPE and RMdSPE for each country (Pakistan, Iran and India) have been reported in the said Table. We have used different starting points to check the predictive power of the proposed models. In particular we have used four different

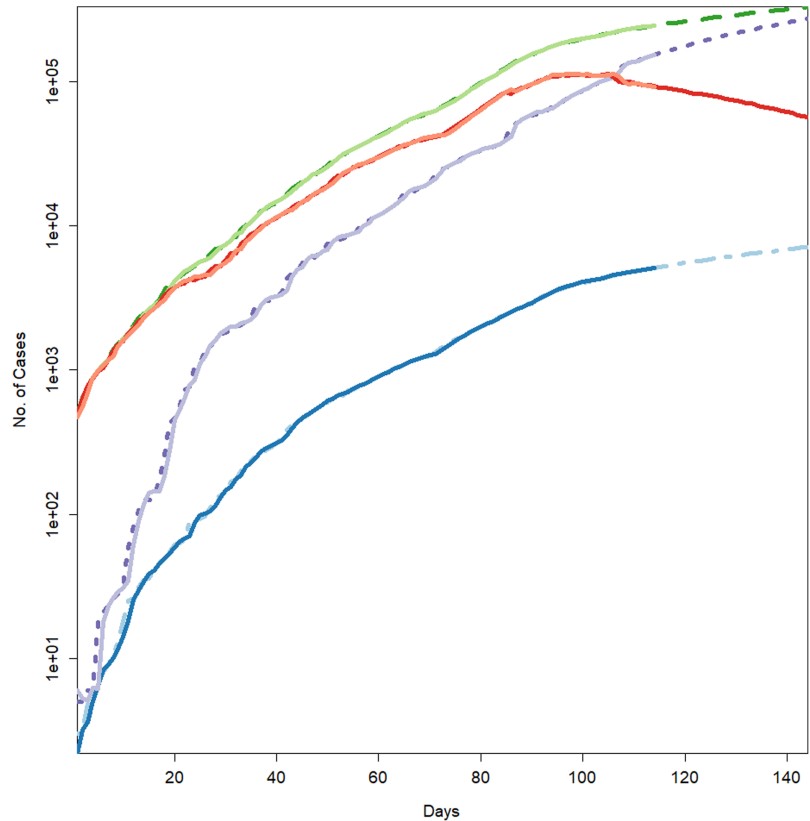

**Figure 2 Forecasts for different parameters of the pandemic using BSTS models.** Each line represents the observed vs fitted and forecast values. The light green represent the forecasts for cumulative number of cases. The dashed dark green line shows the fitted values for cumulative number of cases (which have overlapped the observed numbers of cases given in light green color). The blue line indicates the forecasts for the cumulative number of deaths. The light blue (dotted/dashed) line represents the fitted values for the cumulative number of deaths (which have overlapped the observed numbers of deaths presented in blue color). The light purple line shows the forecasts for the cumulative number of recoveries. The purple line (dotted) depicts the fitted values for the recoveries (which have overlapped the observed numbers of recoveries presented in light purple color). The light maroon line shows the cumulative number of active cases. The dark maroon line represents the fitted values for the active cases (which have overlapped the observed numbers of active cases presented in light maroon color).

starting points which include 90%, 80%, 70% and 60% of the original series. For each staring point the values of RMSPE and RMdSPE are less than 10% (0.10), which indicates the predictive power of the proposed model is good enough. The results in this table also suggest the amounts for RMSPE and RMdSPE are smaller for BSTS models, with few exceptions. Hence, the forecast accuracy of BSTS models is better than that of ARIMA models. The trace plots for the one-step-ahead prediction error variance (PEV), from the MCMC chains, have also been used for the evaluation of the prediction power of the proposed model. The Figs. 5–8 represent the trace plots for PEV regarding cumulative number of cases, deaths, recoveries and active cases in Pakistan, respectively. Each of the
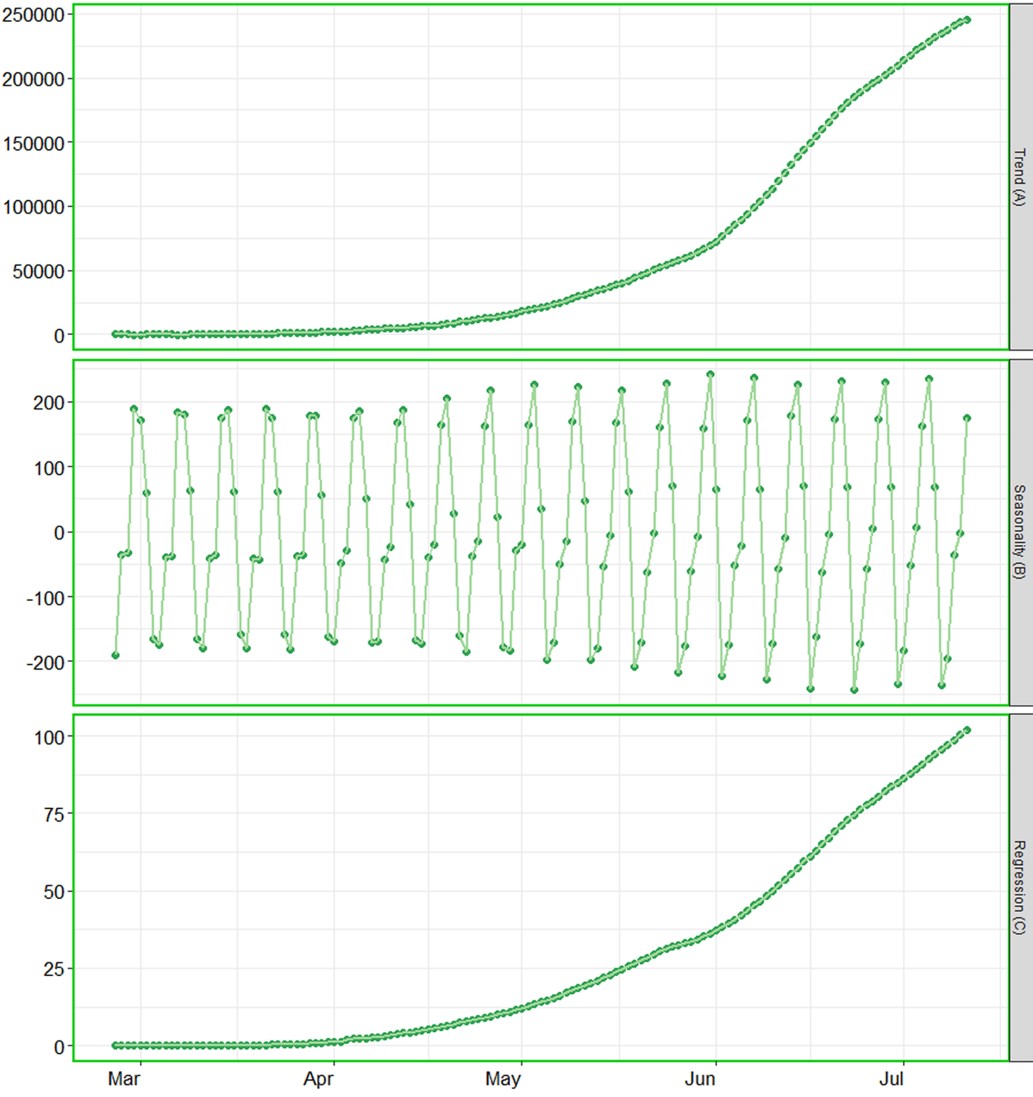

**Figure 3 Analysis of trend, seasonality and trend components.** (A) The contribution of trend component. (B) The contribution of seasonality. (C) The contribution of regression component of the series.

said figures suggested the good convergence of the estimates. Further, QQ plots for the residual terms have been given in Figs. 9–12. In QQ plots, the black line is the reference line for the standard normal distribution and means of MCMC draws have been represented by blue dots. Since the dots are clustered around the straight line, the normality assumption holds well for the residuals. In additions, we have also plotted the posterior distribution of the autocorrelation function (ACF) of residuals using boxplots, for Pakistan, in Figs. 13–16. Since the ACF plots have dampened out at successive lags, so the residual terms are stationary for each variable. Further, the numerical results regarding diagnostic checking of the proposed models have been presented in Table 2.

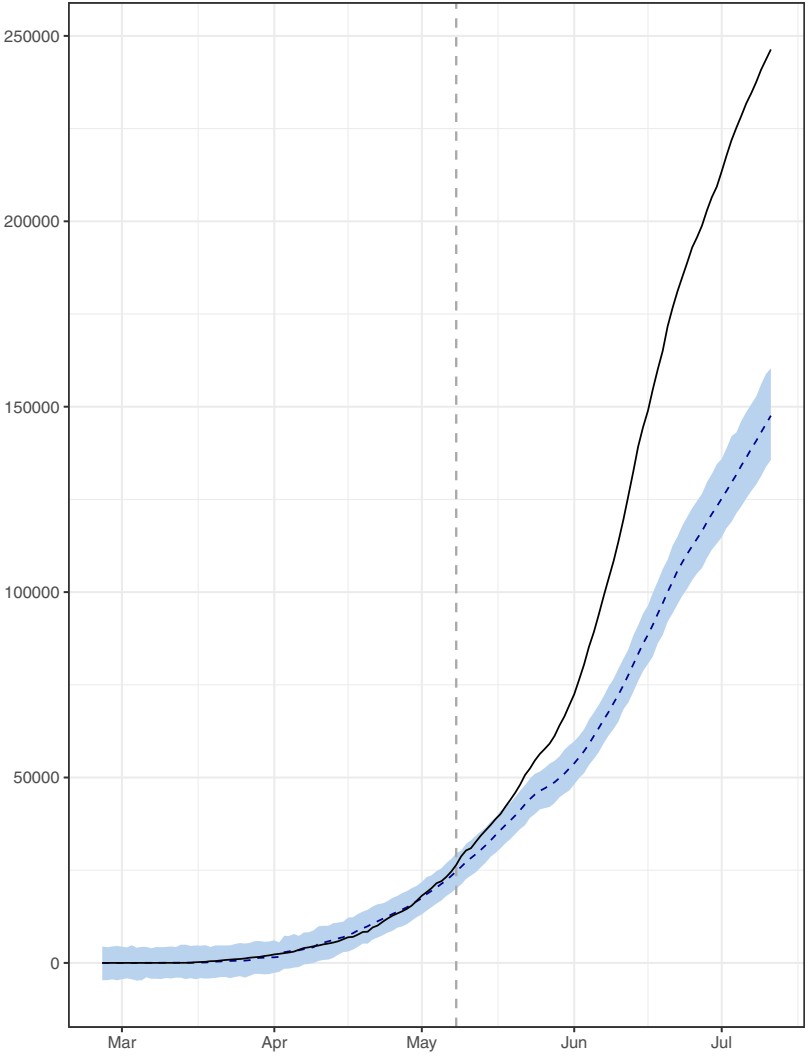

**Figure 4 Analysis of causal impacts of lifting lockdown in the country.** The data points indicate the observed and expected number (hand the lockdown not lifted) of cumulative cases. The black line shows the observed number of cases. The grey line with grey shading indicates the expected number of cases, had the lockdown not lifted.

From these results, it can be seen that $P$-values for the Ljung Box test (Q-statistic) are greater than 0.05 for all variables at different lags. So, we do not have sufficient evidence to assume the residuals as dependent. The calibration for the models using 90%, 80%, 70%, 60% and 50% data has been reported in Figs. 17–21, respectively. From the said figures, it can be seen that actual data are within the corresponding 95% prediction intervals.

Hence, the models can efficiently be used to derive the forecast for the epidemic. Table 3 and Fig. 2 represent the forecasts for various parameters of the epidemic using BSTS models. It should be noted that this figure has logarithmic scale for $y$-axis, so the straight

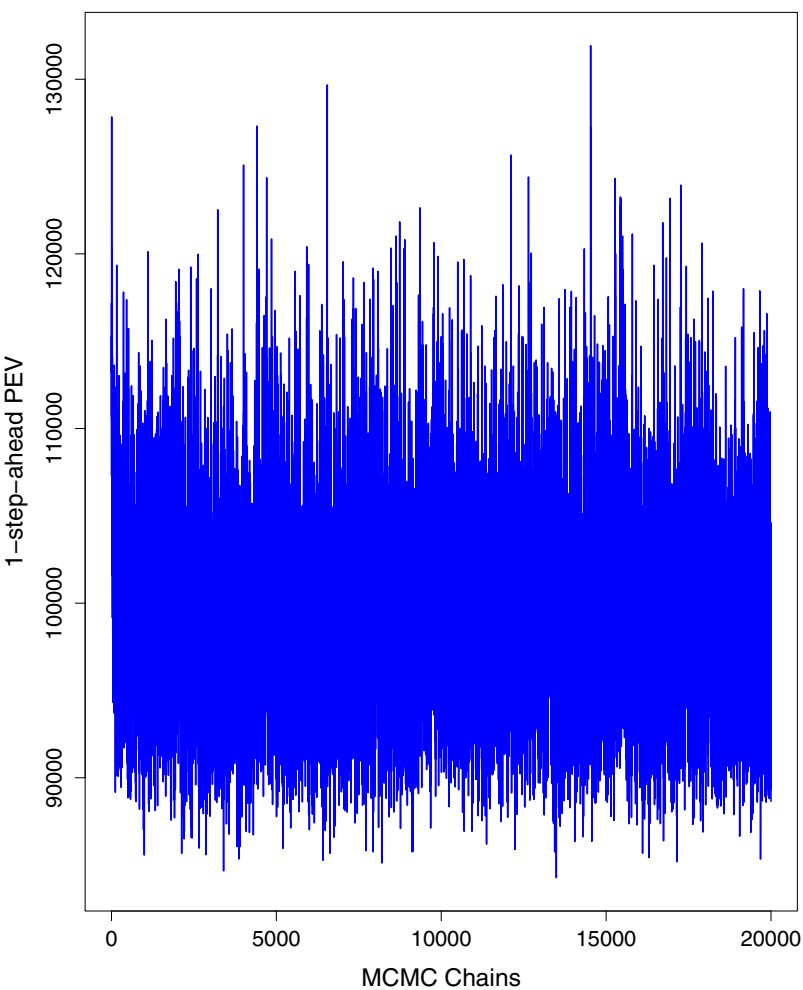

**Figure 5 Trace plots for one-step-ahead prediction error variance (PEV) using MCMC chains for cumulative number of cases in Pakistan.** The blue lines show the average of MCMC draws at each iteration.

line indicate the exponential growth. From this figure, it can be assessed that estimated series is quite close to the observed data (mostly overlapping the observed series), which indicates the model efficiency. Further, the cumulative number of positive cases, recoveries and deaths is expected to increase exponentially during the next 30 days. However, the growth in number of recoveries is expected to be faster than that of confirmed case, which is a good sign for the country. To be more specific, on August 10, 2020, the expected number of positive cases in Pakistan will be 333,308 with 95% prediction interval [275,034–391,077]. Similarly, the number of deaths in the country is expected to reach 7,187 [5,978–8,390] and recoveries may grow to 279,602 [208,420–295,740]. However, it was very encouraging to observe that the active cases are likely to decrease to 63,706 [17,614–95,337], which is lower as compared to current

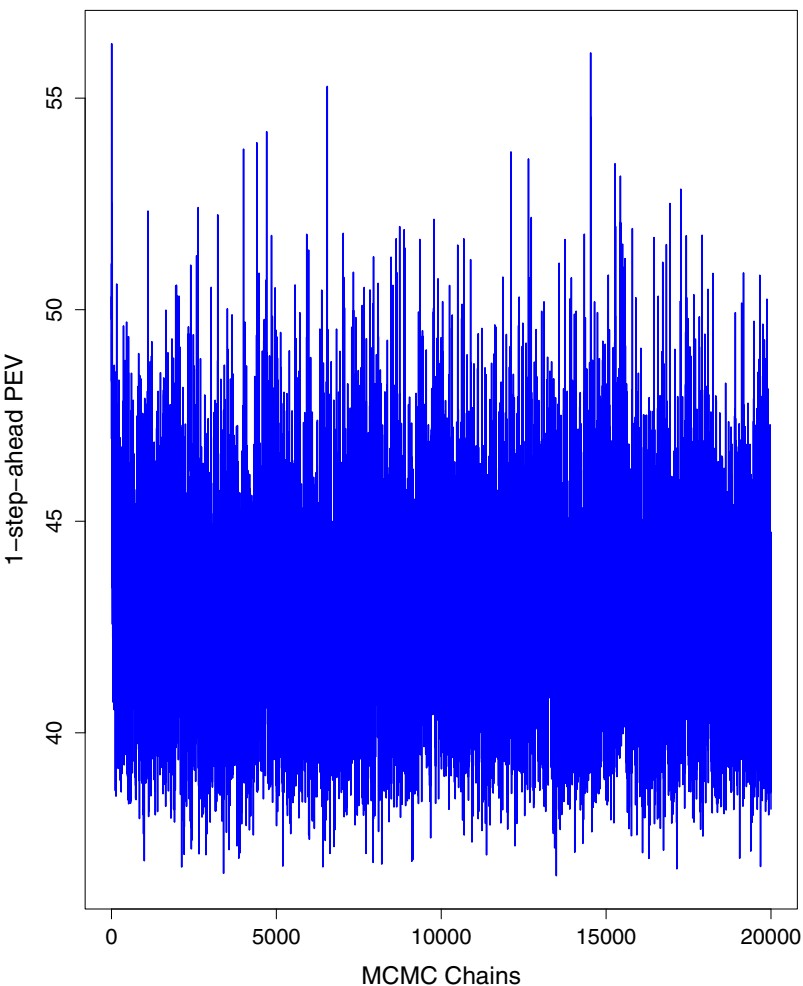

**Figure 6 Trace plots for one-step-ahead prediction error variance (PEV) using MCMC chains for cumulative number of deaths in Pakistan.** The blue lines show the average of MCMC draws at each iteration.

number. This relief is mainly due to rapid growth in the recoveries in the country. The demand for the healthcare facilities is expected to be slightly lower in the next month. The health administration should utilize this opportunity to improve the quality of health services regarding the COVID-19. On the other hand, Iran is expecting 325,391 [283,403–369,613] active cases, 17,458 [15,538–19,476] deaths, 294,493 [243,470–350,203] recoveries and 5,892 [973–73,416] active cases on August 10, 2020. Hence, Iran is expecting quite comprehensive control over the pandemic in the next 30 days. However, the situation of the pandemic in India can be quite serious during next month. Our forecasts for India suggest that the country should expect 1,874,037 [1,538,452–2,393,316] active cases, 39,464 [30,407–48,765] deaths, 1,299,328 [1,091,762–1,618,996] recoveries and 517,153 [390,286–717,715] active cases up to the target date (August 10, 2020).
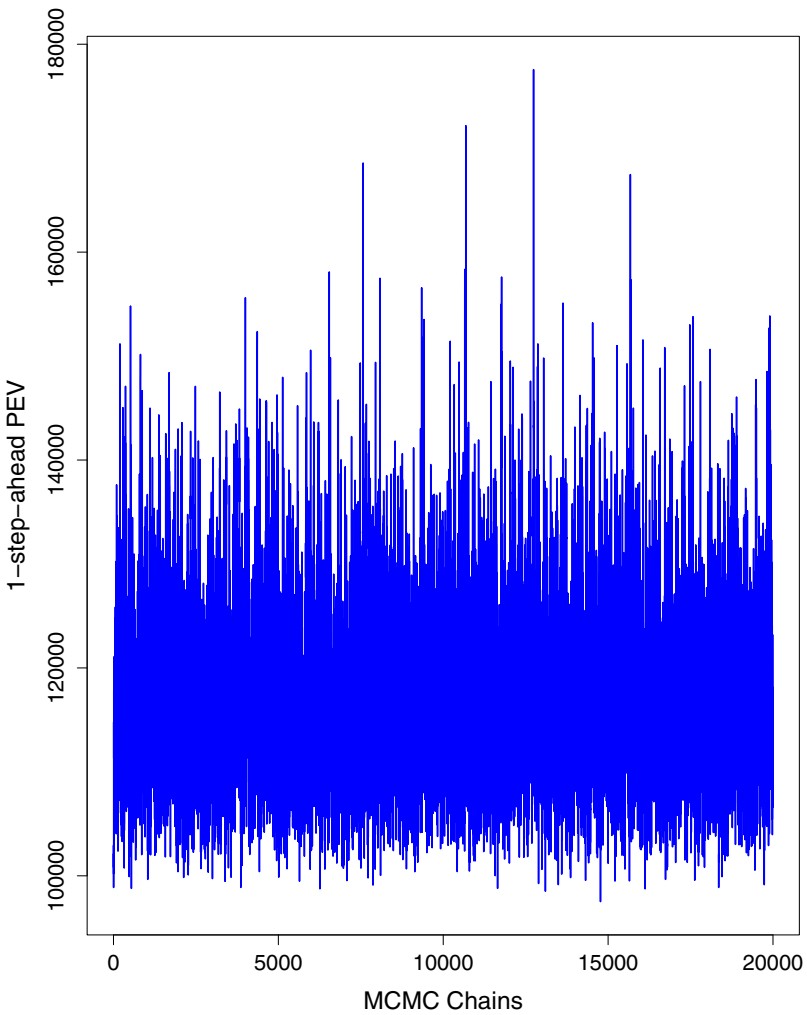

**Figure 7 Trace plots for one-step-ahead prediction error variance (PEV) using MCMC chains for cumulative number of recoveries in Pakistan.** The blue lines show the average of MCMC draws at each iteration.

   The BSTS models also allow us to investigate the patterns of trend, seasonality and regressions individually. We have investigated the contribution of these components for cumulative number of positive cases, as the patterns for the other parameters were alike. As the cumulative number of positive cases depends on the cumulative number of tests, we considered the cumulative number of tests conducted in the country as covariate. The contributions of the said components have been presented in Fig. 3. This figure elucidates that the cumulative number of positive cases is increasing exponentially in the country. The contribution of seasonality is slightly increasing over time. In addition, the contribution of regression component is also has an upward trend, this is mainly due to increase in rate of confirmed cases.

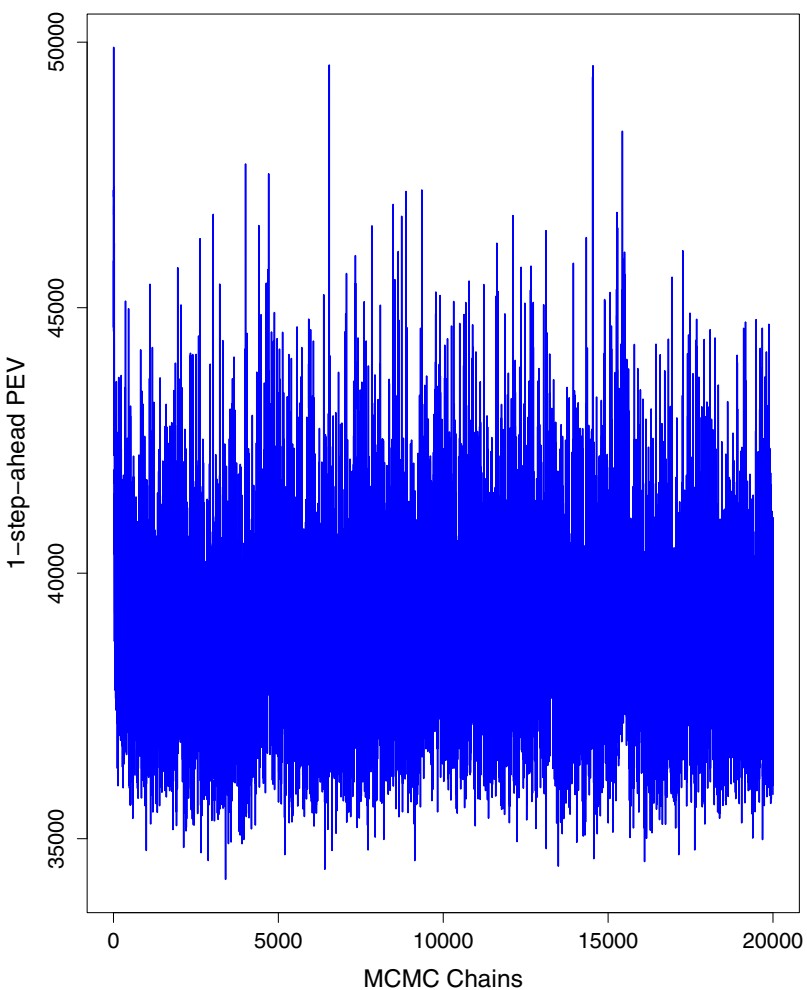

**Figure 8 Trace plots for one-step-ahead prediction error variance (PEV) using MCMC chains for number of active cases in Pakistan.** The blue lines show the average of MCMC draws at each iteration.

The causal impact of lifting the lockdown in the country has also been discussed by conducting intervention analysis using BSTS models. The results have been given in Fig. 4. This figure simply indicates the significant increase in pace of the outbreak (in black lines) as compared to expected, had the lockdown not lifted (in grey shade).
The probability of causal impact was 0.9989, which was quite high indicating that the lifting the lockdown has increased the outbreak of the epidemic significantly. On July 11, 2020, the total number of confirmed cases was 246,351; however under lockdown the number of cases would have been 147,583 with 95% interval [135,333–160,807]. So, there is an absolute increase of 98,768 confirmed cases with 95% interval [85,544–111,018]. Hence, the country has conceded 56% increase (in the confirmed cases) than the expected, if the lockdown would have continued. In addition, during the
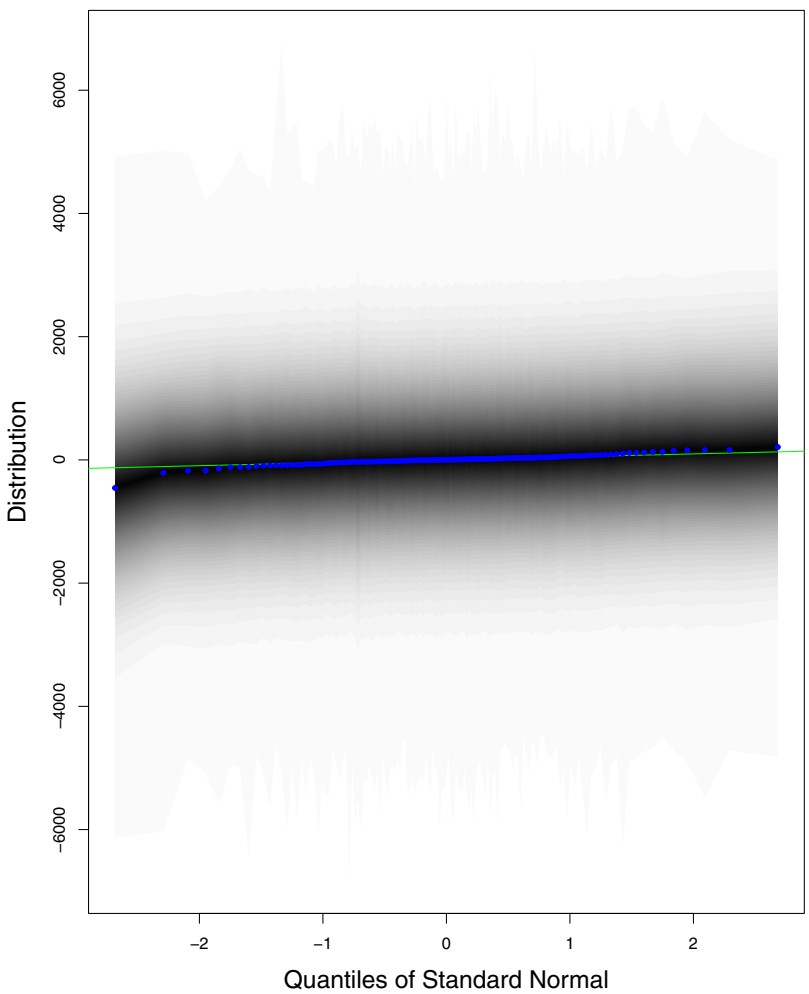

**Figure 9 QQ plots for the residual terms for cumulative number of cases in Pakistan.** The black line is the reference line for the standard normal distribution and means of MCMC draws have been represented by blue dots.               

lockdown, the average rate of positive cases per 100 tests was 8.68%, while for the post lockdown period this rate stands at 13.21%. Similarly, the rate of deaths per 100 positive cases, for the lockdown period, was 1.46% which increased to 2.06% in the after the lockdown. On the whole, the ending lockdown has significantly increased the load on the Pakistani healthcare system. It is also a truth that the country cannot afford a prolonged lockdown. However, identifying the hotspots and enforcing lockdown in those areas may help. In addition, the serious effort to induce the people to abide by the SOPs in the country is fundamental.

The results from other studies of this nature are quite compatible. For example, the studies conducted by *Li et al. (2020)*, *Majeed, Adeleke & Popoola (2020)*, *Tomar & Gupta (2020)*, *Perc et al. (2020)*, *Fanelli & Piazza (2020)*, and *Moftakhar, Seif & Safe (2020)*, also

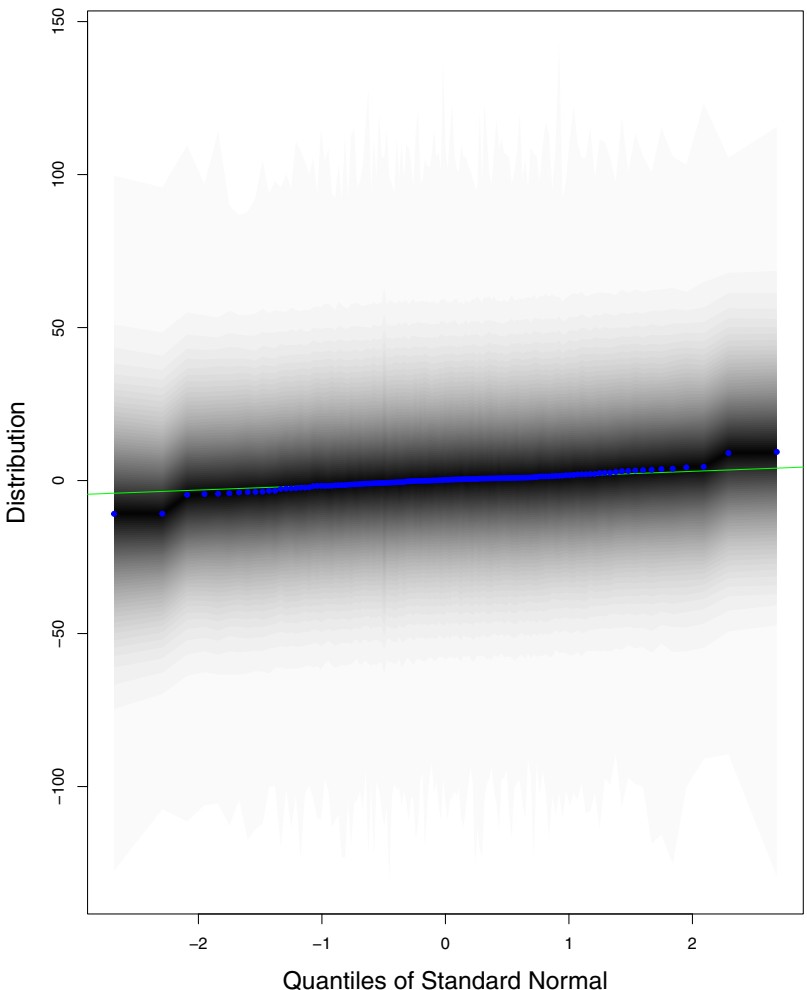

**Figure 10 QQ plots for the residual terms for cumulative number of deaths in Pakistan.** The black line is the reference line for the standard normal distribution and means of MCMC draws have been represented by blue dots.

indicated a rapid future growth in the pandemic in respective countries. Our results are also comparable with the earlier study carried out in Pakistan (*Yousaf et al., 2020*). However, this study did not cover the impact lockdown in the country. In addition, we have obtained these forecasts using a more flexible model.

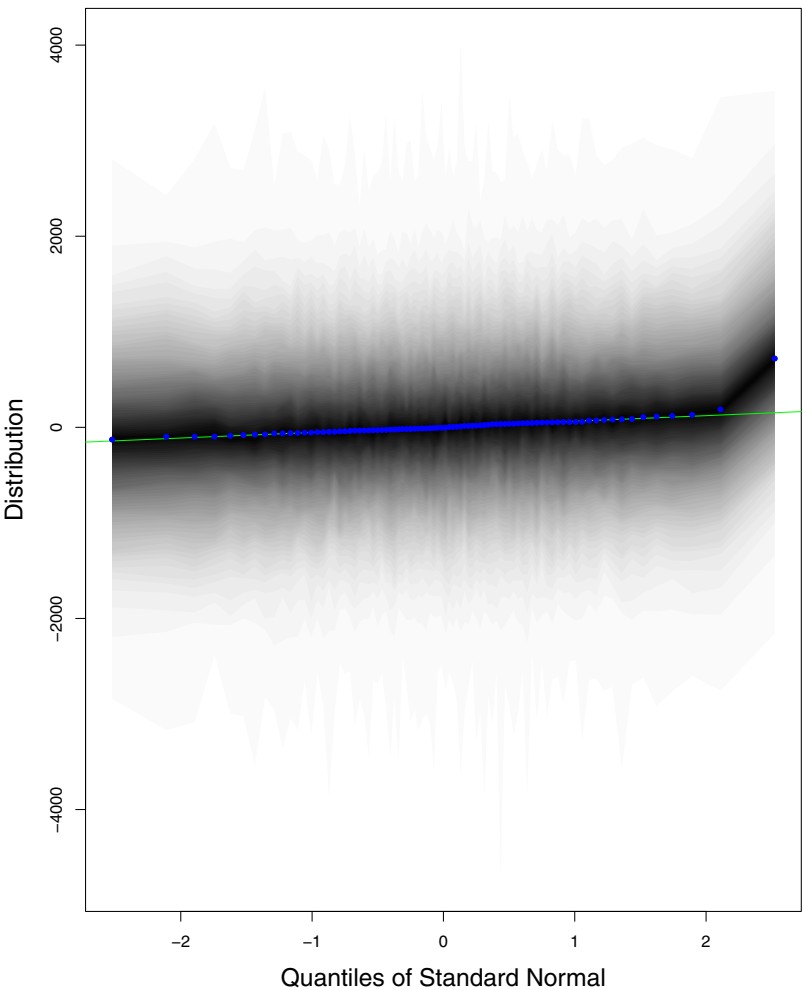

**Figure 11 QQ plots for the residual terms for cumulative number of recoveries in Pakistan.** The black line is the reference line for the standard normal distribution and means of MCMC draws have been represented by blue dots.

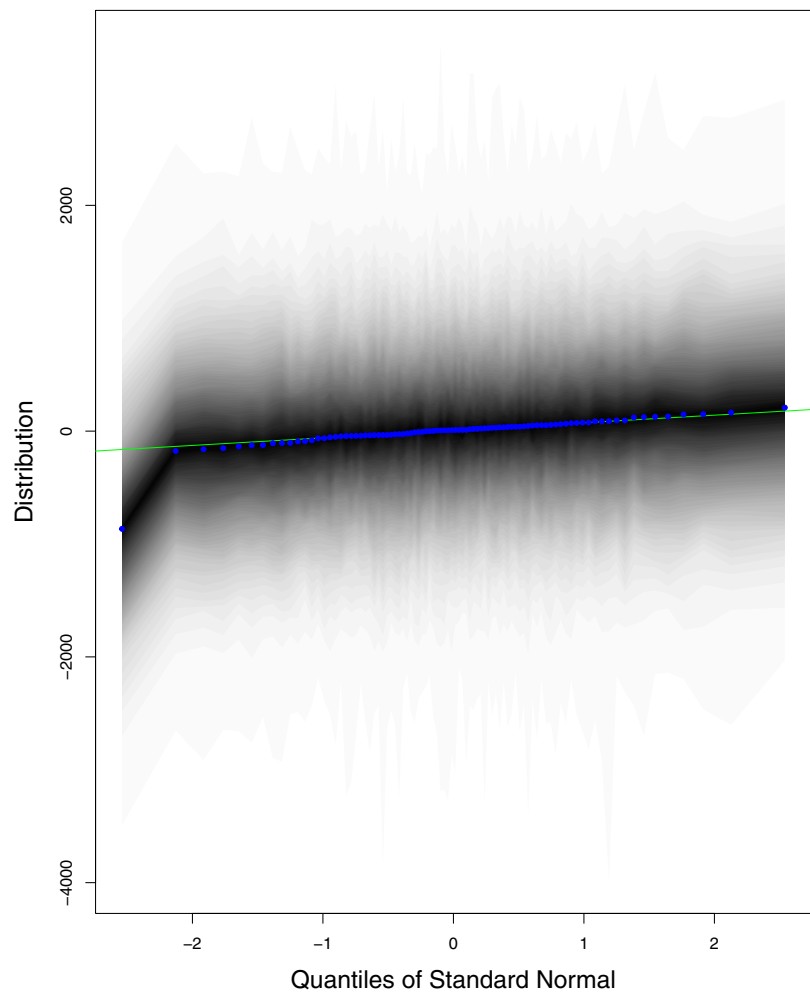

**Figure 12 QQ plots for the residual terms for number of active cases in Pakistan.** The black line is the reference line for the standard normal distribution and means of MCMC draws have been represented by blue dots.
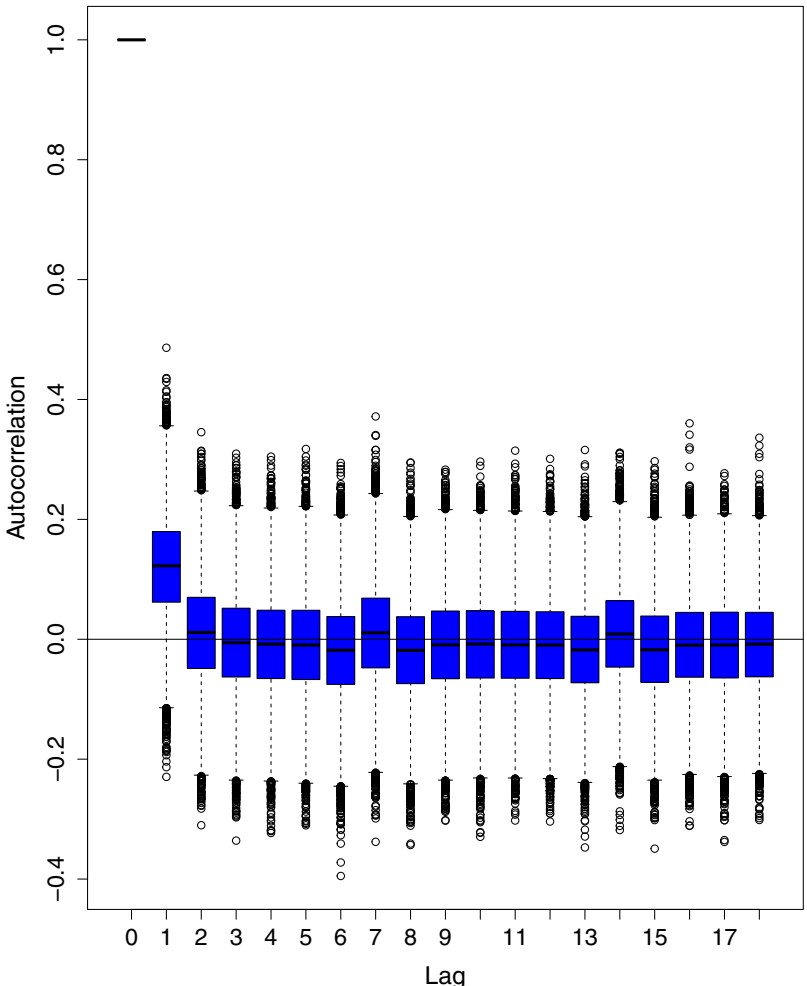

**Figure 13 Posterior distribution of the autocorrelation function (ACF) of residuals using boxplots for cumulative number of cases in Pakistan.** The blue boxes represent side-by-side box plots.

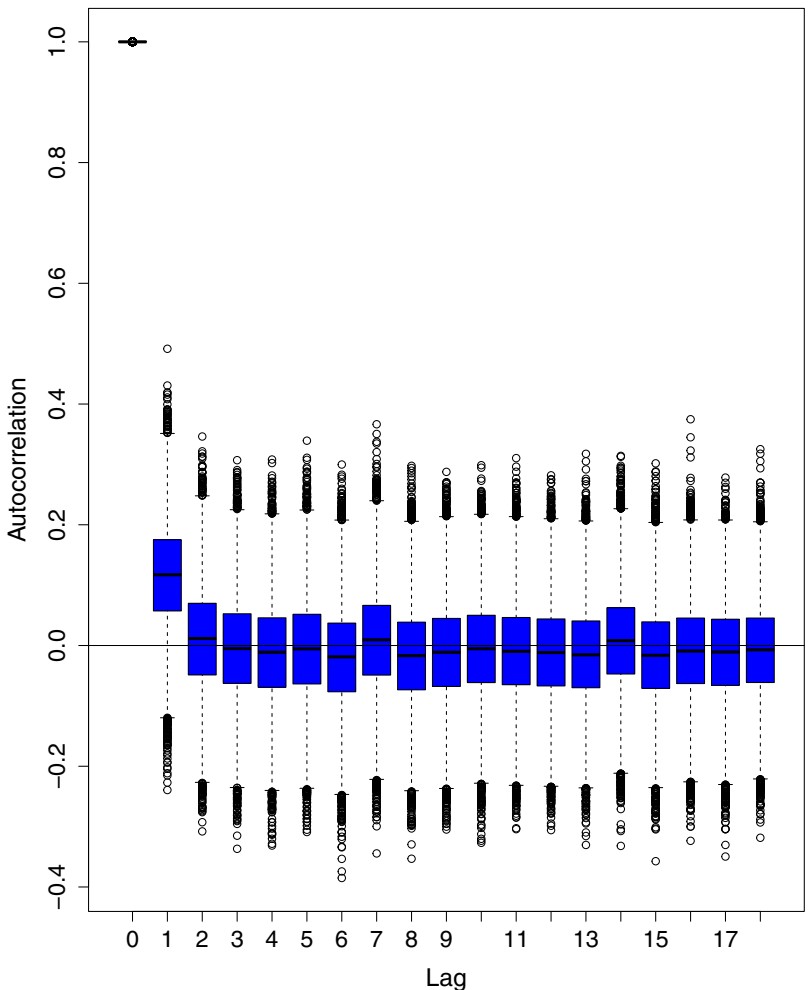

**Figure 14 Posterior distribution of the autocorrelation function (ACF) of residuals using boxplots for cumulative number of deaths in Pakistan.** The blue boxes represent side-by-side box plots.

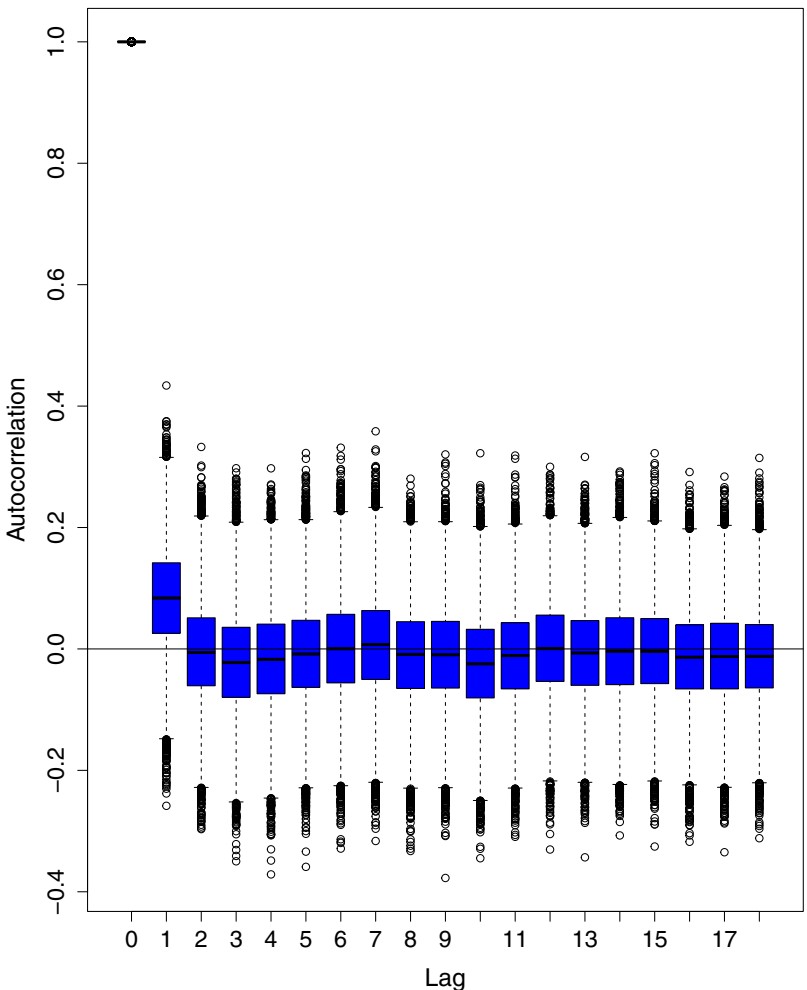

**Figure 15 Posterior distribution of the autocorrelation function (ACF) of residuals using boxplots for cumulative number of recoveries in Pakistan.** The blue boxes represent side-by-side box plots.

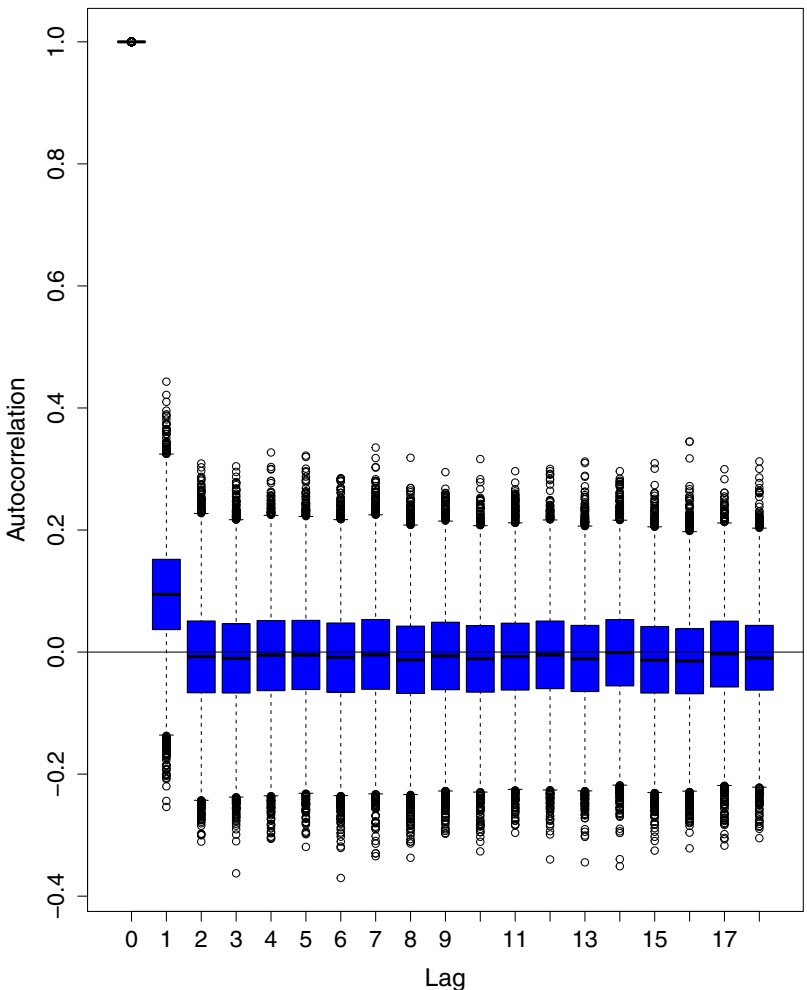

**Figure 16 Posterior distribution of the autocorrelation function (ACF) of residuals using boxplots for number of active cases in Pakistan.** The blue boxes represent side-by-side box plots.

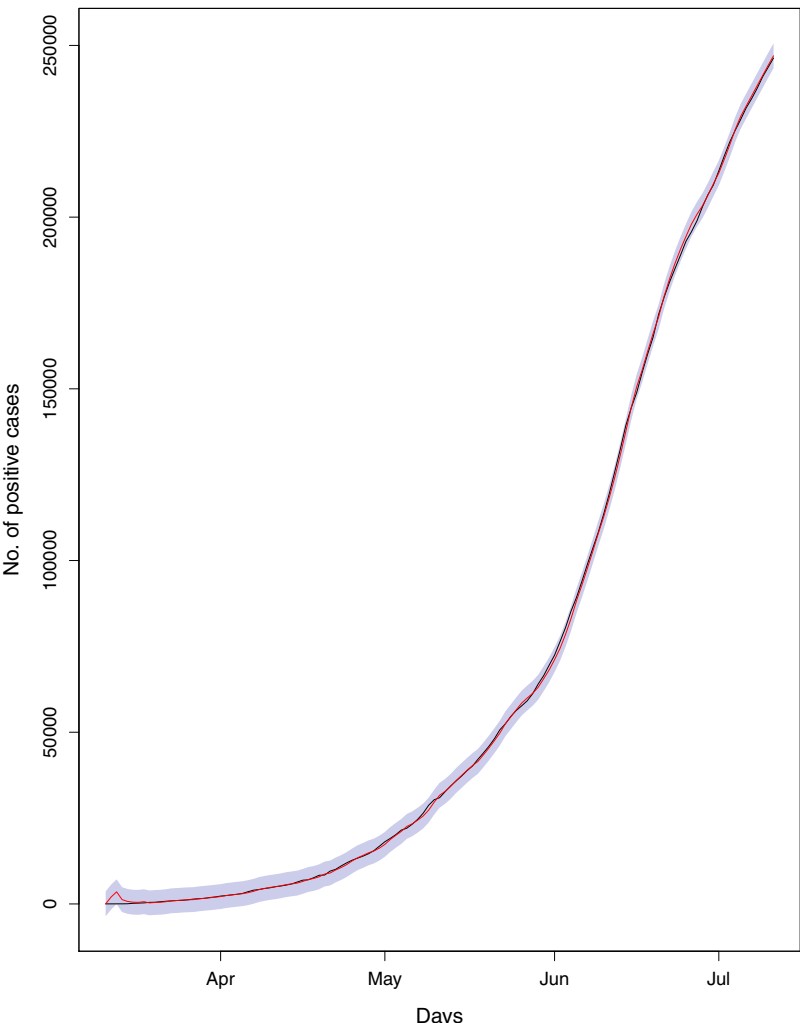

**Figure 17 Comparison of true vs fitted series using 90% of the true dataset.** Black line shows the actual series. Maroon line shows fitted series and shaded (light maroon) area shows the 95% prediction interval.

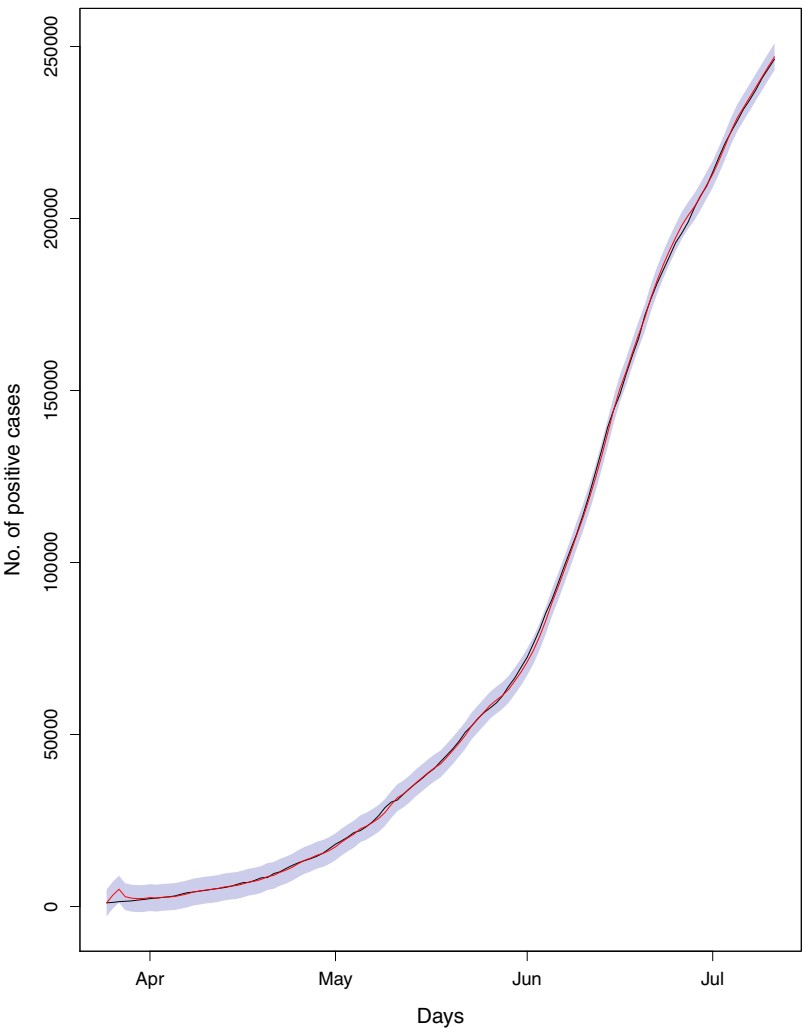

**Figure 18 Comparison of true vs fitted series using 80% of the true dataset.** Black line shows the actual series. Maroon line shows fitted series and shaded (light maroon) area shows the 95% prediction interval.

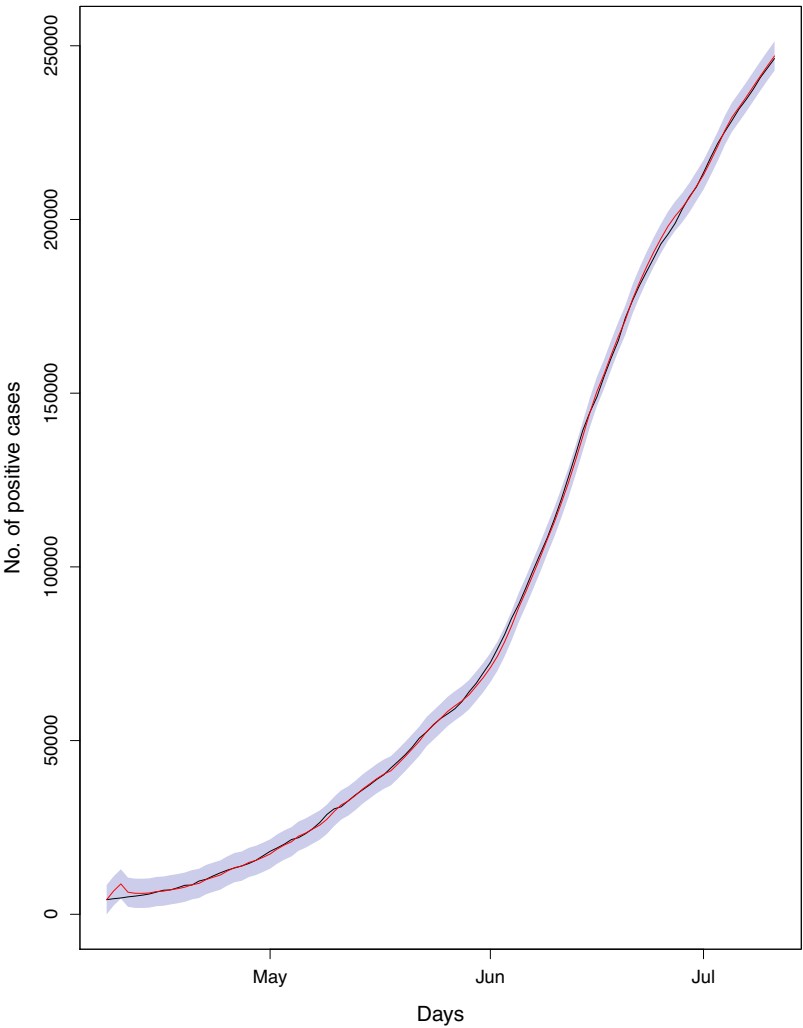

**Figure 19 Comparison of true vs fitted series using 70% of the true dataset.** The black line shows the actual series. Maroon line shows fitted series and shaded (light maroon) area shows the 95% prediction interval.

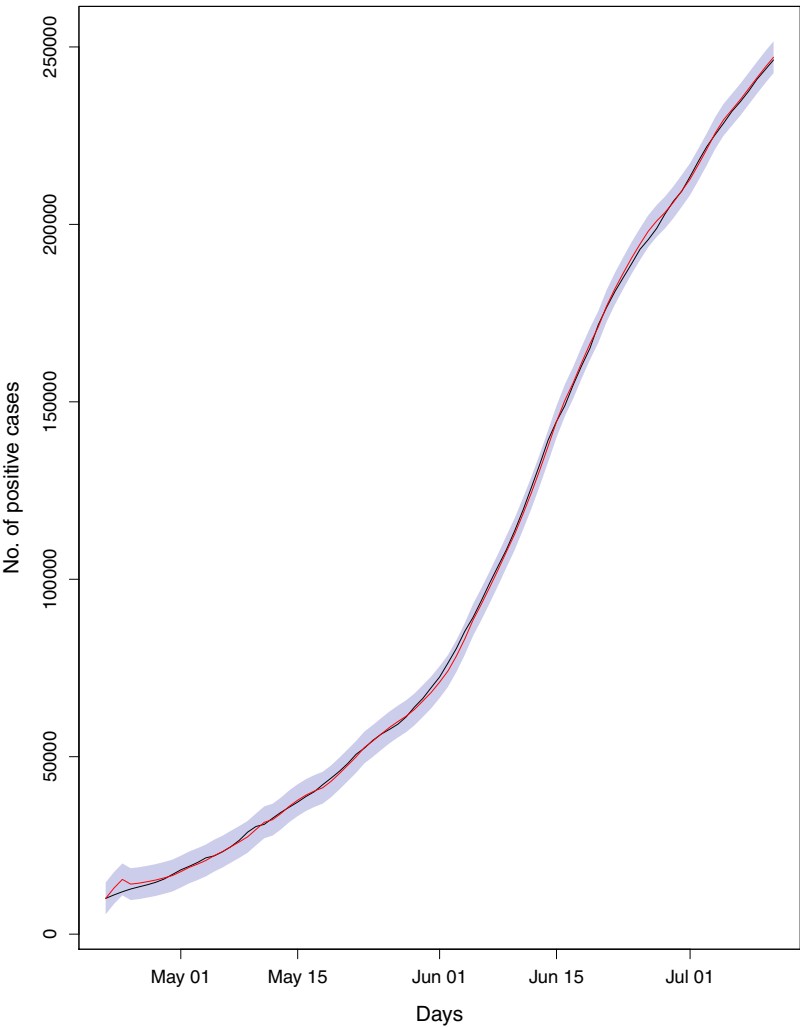

**Figure 20 Comparison of true vs fitted series using 60% of the true dataset.** The black line shows the actual series. The maroon line shows fitted series and shaded (light maroon) area shows the 95% prediction interval.

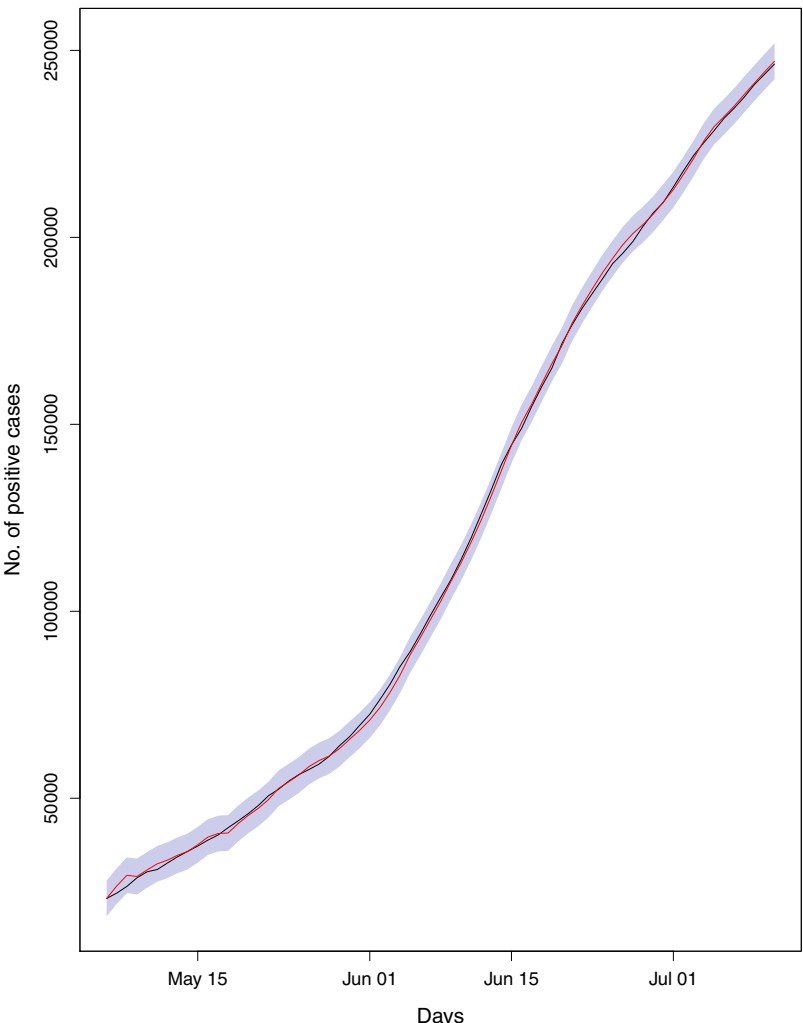

**Figure 21 Comparison of true vs fitted series using 50% of the true dataset.** The black line shows the actual series. The maroon line shows fitted series and shaded (light maroon) area shows the 95% prediction interval.

## CONCLUSIONS

The study has been conducted to investigate the different parameters of COVID-19 in Pakistan. The study suggests that the lifting of lockdown has increased the speed of the pandemic in the country. The number of positive cases is directly proportional to the number of tests carried out in the country. The total number of positive cases and deaths is expected to grow exponentially. However, the trajectory of recoveries is likely to be higher than that for confirmed cases. Therefore, the expected number of active cases is likely to decrease in the next month. We suggest that this is the right time where concerned authorities may put more efforts to further minimize the active patients in the country. If it happened, there are expectations that country will see the contraction in the outbreak of this pandemic. It should also be noted that Pakistan cannot afford the complete lockdown due to its frail economy; hence, we cannot suggest the complete lockdown in the country again. However, properly identifying hotspots and enforcing lockdown partially may help. The government also needs to follow a strict tracking, tracing and quarantine strategy. More importantly, all possible efforts and resources should be mobilized to improve the healthcare facilities to meet the needs of the coming month. Similarly, the overall situation of the pandemic in Iran is expected to be quite in control in the next month. However, India is expecting a drastic increase in number of patients, deaths and active cases in the coming month. Though, the number of recovered patients is expected to grow exponentially, the impact of this growth will be seized by the fast growing cases and deaths.

Our study has some limitations. Firstly, the data may be underreported, as the sufficient random testing has not been considered in the country, and not all the patients report themselves to health officials with a fear of staying 14 days in quarantine. Secondly, the forecasts are based on the assumption that the current trends will follow in the next month, the forecasts can be misleading if this assumption is violated. Thirdly, the study is lacking in investigating the risk factors as the data regarding the demography and social networks of the patients were not available.

### Funding
No funding was received.

### Competing Interests
The authors declare that there is no competing interests

### Author Contributions
- Navid Feroze conceived and designed the experiments, analyzed the data, prepared figures and/or tables, and approved the final draft.
- Kamran Abbas conceived and designed the experiments, prepared figures and/or tables, and approved the final draft.

- Farzana Noor performed the experiments, analyzed the data, authored or reviewed drafts of the paper, and approved the final draft.
- Amjad Ali performed the experiments, authored or reviewed drafts of the paper, and approved the final draft.

## Data Availability

The data and the R code are available in the Supplemental Files.

## Supplemental Information

Supplemental information for this article can be found online at http://dx.doi.org/10.7717/peerj.11537#supplemental-information.

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
