# Peer review of "Analysis and forecasts for trends of COVID-19 in Pakistan using Bayesian models"

_PeerJ, doi:10.7717/peerj.11537_

## Round 0.1 · original submission · Major Revisions

Dear authors,

Three experts have assessed your work and they have found scientific merit in it. However, there are some issues that you must address in a revised version of the text.

Best regards,
Dr Palazón-Bru

Reviewer 1 ·

Basic reporting

The paper seems to contains interesting material.
The state-of-the-art discussion must be improved.
More explanations are needed in several parts of the work.

Experimental design

The paper seems technical sound but more explanation of the goal and methods are required.

Validity of the findings

The degree of novelty is not high however the paper seems technical sound and the results are interesting.

Additional comments

The paper seems to contains interesting material. More explanations are needed in several parts of the work. See my comments below.

- Please, write mathematically and properly the used ARIMA model, defining all the variables, distributions, constants etc. Clarifying what is known and what is unknown. Please, clarify the meaning of each variable which variable represents which physical amount etc.

- Please, explain better the contents of the Tables in the text and in their caption.

- Please, clarify the degree of novelty and the your contribution in the introduction in a better way.

- The state-of-the-art discussion must be improved. Please, clarify which method you use to make the inference as MCMC or Adaptive Importance Sampling (AIS) methods. Even if you do not use AIS or MCMC it is important to clarify that in a general Bayesian inference they are needed. For instance, consider the important and relevant review papers

M. Bugallo et al, "Adaptive Importance Sampling in Signal Processing", Digital Signal Processing, Volume 47, Pages: 36-49, 2015,

L. Martino, "A Review of Multiple Try MCMC algorithms for Signal Processing", Digital Signal Processing, Volume 75, Pages: 134-152, 2018,

W.J. Fitzgerald, "Markov chain Monte Carlo methods with applications to signal processing", Signal Processing, Volume 81, Issue 1, Pages 3-18, 2001.

This discussion can improve the number of interested readers.

·

Basic reporting

The paper was easy to read and I appreciated the brevity.

Code and data were available, I was easily able to reproduce the findings and figures. Good job.

## Major issues

Currently the paper is a bit hard to understand on its own as I don’t think BSTS are common enough model to be known to readers (I never heard of it myself before). Methods should include a brief description of the structure and basic assumptions of BSTS models and the specifics of the model you actually used (the components of the models, the number of seasons, the fact that you modelled each time series individually, …). A short argument why you expect the assumptions of the model to not be severely violated by the data could also be helpful.

Similarly, an overview of the assumptions and mechanism of CausalImpact should be mentioned.

You should probably also cite and discuss another modelling effort for Pakistan by Khan et al 2020: Modelling and forecasting of new cases, deaths and recover cases of COVID-19 by using Vector Autoregressive model in Pakistan https://doi.org/10.1016/j.chaos.2020.110189

## Minor issues

I find the Figure 1 too crowded - pure line plot would in my view be preferable.

The color palettes are in my view not very nice and also not accessible to people with impaired color perception - red-green contrasts are particularly hard to perceive for a lot of people. Consider using palettes from R packages viridis, colorbrewer or scico which have been assessed for accessibility (and in my view look better). Also use line type, not only colour to distinguish the individual series.

I would expect to have figures and tables embedded in text for submission version, I understand some journals still require this way of submitting, but I don’t think PeerJ does and it makes my job as a reviewer unnecessarily harder.

Minor linguistic suggestion can be found in the attached PDF.

Experimental design

The paper tackles an important problem and I believe Pakistan deserves the best predictions it can get. I am also a big supporter of using Bayesian statistics and I am glad authors chose to use it.

Generally, I think the authors need to decide whether the aim is to provide high-quality predictions for Pakistan (and thus multiple models with very different assumptions should be fit to the Pakistan data to assess uncertainty from model choice), or whether the aim is to evaluate to what extent BSTS are a sensible tool for Covid/disease prediction in general (and thus datasets from more countries should be used).

One possible model to use/compare against would be the epidemia package (https://imperialcollegelondon.github.io/epidemia/) or other models mentioned covid19forecasthub.org

The biggest risk with the current design is in my view that the model doesn’t use the structure of the data (e.g. that all cases eventually become death or recoveries, all deaths/recoveries were once cases, the number of cases is limited by the total population, …). The epidemia package uses this structure and would thus be an interesting point of comparison (though I in no way want to force the authors to use this specific package).

Similarly, most successful models I know of do not focus on cumulative cases, but on new cases, as the cumulative data can be “too smooth” and thus underestimate the uncertainty. Comparing to/pairing with a method that focuses on new cases (or generally the time derivative of the time series) would be beneficial (but I don’t think it is required).

Evaluating predictive ability for just one time point is quite a weak check. Testing prediction should be done starting from multiple historical time points (preferably even a “sliding window” approach where say each data point is predicted from data say 5, 10 and 30 days prior, but that’s not completely necessary).

I also do take issue with the metrics used: while RMSE and others can be useful, in a Bayesian and decision making setting, it is important to also check if the uncertainty is correctly quantified. The calibration of the model is important (i.e. across multiple starting points, do X% of actual data fit within the X% prediction interval for various values of X?).

On the other hand I do NOT think BSTS need to be shown to be a great method for the manuscript to be publishable. If a revised version of the paper provides a set of carefully checked predictions for Pakistan and/or a good evaluation of BSTS across multiple Covid datasets, I don’t care if the conclusions are not favorable to BSTS and would happily recommend the publication for acceptance.

Validity of the findings

The two following issues are in my view major and need to be addressed.

1) Using any MCMC algorithm, it is important to check that the MCMC chain converged. Multiple chains should be compared to assess if the whole posterior is explored. Unfortunately this has not been done. Some basic tools for convergence checks for a Gibbs sampler would be traceplots, R-hat, and Effective sample size - see e.g. Vehtari et al. 2020, http://doi.org/10.1214/20-BA1221

And actually some of the parameters, especially “sigma.trend.level” seems to not have converged. Here’s my code I used to check convergence for some of the parameters, using RStan’s implementation of R-hat and ESS. Note that R-hat should roughly be < 1.01 and bulk ESS of at least 50 and usually much more is needed to accurately estimate a quantity (ess_tail should also be used and be high enough if tail-quantities, like the 95% interval are of interest).

fits <- list()
n_chains <- 4
for(i in 1:n_chains) {
fits[[i]] <- bsts(ts, state.specification = ss, niter = 500, ping=0, seed=2016 + (i - 1) * 157555)
}

for(par_name in c("sigma.obs", "sigma.trend.level", "sigma.trend.slope", "sigma.seasonal.2" ) ) {
draws <- matrix(nrow = length(fits[[1]][[par_name]]) - burn, ncol = n_chains)
for(i in 1:n_chains) {
draws[, i] <- fits[][[par_name]][(burn + 1) : length(fits[][[par_name]])]
}
cat(par_name, ", Rhat: ", rstan::Rhat(draws), ", ESS: ", rstan::ess_bulk(draws), "\n")
}

This should be done for all parameters (and preferably a bit neater, the code is a bit hacky, sorry).

The traceplots also do show that the chains actually explore a slightly different part of the posterior.

Thus more iterations, longer burnin, and possibly thinning need to be introduced to make the computation trustworthy.


2) I also find it very weird that different number of seasons is used for different pandemic counts. Seasons are of length 1 (days), but why are 4 or 2 day trends chosen? I would expect the pandemic to have a weekly pattern, but why should there be a 2-day or 4-day pattern? How were these parameters chosen?


## Additional note

Comparing to actual numbers for their target date (August 10th) the reality is on the lower end of the confidence intervals and the number of active cases (17 799) falls quite outside the 95% prediction interval [22,568;102,717]. I suspect this is because the model seems to fit a continued exponential growth, which seems to have not happened. I don’t think it is reasonable to fail the authors for not predicting accurately, as this is very hard. But - if I understand the model correctly, I think the structure of the model may have been unable to model a plateau for the number of total cases, which is in my opinion an important oversight.

Additional comments

I wish you best of luck with modelling and with the overall pandemic situation in general. I hope you are safe.

Reviewer 3 ·

Basic reporting

The authors proposed a computational model for forecasting the COVID-19 trends in Pakistan, thus I think the prediction here is not "pattern" like the authors mentioned in the title. It should be "trends" or "outbreak"?

There are a lot of grammatical errors and typos in this manuscript. The authors should re-check and revise carefully. The use of English language should be also improved significantly. For example, some errors are as follows:
- the first sentence of abstract: Though Pakistan is among the countries where the COVID-19 entered quite later, but currently it is on full flow ...
- To tackle with challenges in the coming months ...
- And timeline of these forecasts have passed now.
- due to which the active number of cases are expected to decrease during ...
- These are quite efficient measure of forecast accuracy of a model.
- ...

There are missing literature review on COVID-19 outbreak prediction in other countries.

Experimental design

Methodology was poorly written. Now it contains two short paragraphs with few information. With these information, the readers could not replicate the methods.

Forecast/prediction model is a well-known problem and there are a lot of models have been developed on this purpose, from machine learning (i.e., PMID: 31055655, PMID: 27475771), or deep learning (PMID: 28643394, PMID: 31921391). Therefore, the authors should show more detail on their model information. Also, it needs more references to attract broader readership in this part.

How did the authors select the optimal hyperparameters of their model?

Validity of the findings

The application is narrow. The authors only focused on COVID-19 outbreak in a specific country (Pakistan). Even in Pakistan, the authors have not validated the model on an unseen data.

I'm wondering whether the model could be efficient in predicting the COVID outbreak in another dataset from the other countries. The authors should implement and discuss this point. It strongly support the hypothesis to provide a useful method in this problem.

The authors should compare the performance results with the previous works on the same dataset.

Additional comments

No comment.

---

## Round 0.2 · Minor Revisions

Still pending some minor changes which you should address in a new revised version of the text.

Reviewer 1 ·

Basic reporting

The paper has been improved, however there are still some points that are clear.

For instance, the model ARIMA in Eq. (1) is instead an ARMA model? where is "d" in Eq. (1)? I believe that you are using and ARMA.

Experimental design

The paper seems technically sound and experiments are good enough.

Validity of the findings

The paper seems technically sound and experiments are good enough.

Additional comments

The paper has been improved, however there are still some points that are clear.

For instance, the model ARIMA in Eq. (1) is instead an ARMA model? where is "d" in Eq. (1)? I believe that you are using and ARMA.

·

Basic reporting

The reporting has been substantially improved from previous version, I have no further bigger requirements.

Minor points:
The authors have not improved the palettes used in their plots and I still find the plots to be not very accessible to people with impaired vision, neither pretty.

Experimental design

Authors have expanded their evaluation and I have no bigger remaining issues.

Minor points:
Authors have not reported calibration, as I asked to in my previous review. I don't think it is strictly necessary, but it would still be useful. (my older suggestion: "The calibration of the model is important (i.e. across multiple starting points, do X% of actual data fit within the X% prediction interval for various values of X?)."). Note that low prediction error does not imply good calibration (or vice versa).

Validity of the findings

Only important problem I noticed:

"From these results, it can be seen that P-values for the Ljung Box test(Q-statistic) are greater than 0.05 for all variables at different lags. Therefore, the residuals from the proposed models are white noise."

This is not a correct interpretation of p-values. The test indicates that the residuals are consistent with white noise (as far as the test is able to discern), but they still could be structure that just is not discernible with the current data.

Additional comments

The article was substantially improved. Not all of my concerns were addressed, but I don't find any value in further gatekeeping the publication.

Reviewer 3 ·

Basic reporting

no comment

Experimental design

no comment

Validity of the findings

no comment

Additional comments

My previous comments have been addressed.

---

## Round 0.3 · accepted · Accept

All the reviewers' concerns have been correctly addressed.

Reviewer 1 ·

Basic reporting

The paper has been improved following my suggestions.
I believe that is ready for publication.

Experimental design

The paper has been improved following my suggestions.
I believe that is ready for publication.

Validity of the findings

ok

Additional comments

The paper has been improved following my suggestions.
I believe that is ready for publication in its current form.

·

Basic reporting

The manuscript was OK in the previous version, it is also OK after the minor changes.

Experimental design

The manuscript was OK in the previous version, it is also OK after the minor changes.

Validity of the findings

The manuscript was OK in the previous version, it is also OK after the minor changes.

Additional comments

The manuscript was OK in the previous version, it is also OK after the minor changes.